# Interactive Effects of Flooding Duration and Sediment Texture on the Growth and Adaptation of Three Plant Species in the Poyang Lake Wetland

**DOI:** 10.3390/biology12070944

**Published:** 2023-07-01

**Authors:** Ying Liu, Jie Li, Yizhen Liu, Liang He, Shanshan Yang, Huiying Gong, Ruixin Xu, Xingzi Yao, Gang Ge

**Affiliations:** 1Key Laboratory of Poyang Lake Environment and Resource Utilization Ministry of Education, Nanchang University, Nanchang 330031, China; yingliu@ncu.edu.cn (Y.L.); lianghe@ncu.edu.cn (L.H.); econcu@163.com (G.G.); 2Institute of Life Science, Nanchang University, Nanchang 330031, China; 3School of Life Science, Nanchang University, Nanchang 330031, China; leecan66@163.com (J.L.); ncuskyangshanshan@163.com (S.Y.); ncuskgonghuiying@163.com (H.G.); ncuskxuruixin@163.com (R.X.); ncuskyaoxingzi@163.com (X.Y.)

**Keywords:** flooding duration, sediment texture, wetland plant, growth, response

## Abstract

**Simple Summary:**

Flooding duration and sediment texture were designed in the study to consider the response of hygrophilous plants to environmental dynamics, which involved three plant species commonly found in the Poyang Lake wetland. The results highlight that sediment texture plays a crucial role in the growth and adaptation of hygrophilous plants, although it exhibited a weaker effect than flooding duration. Additionally, sediment texture mediates flooding to affect plant stressing response and survival. Moreover, sediment texture forms interactive effects with flooding duration and has direct influences on hygrophilous plants. Our study helps provide theoretical insights from a more scientific perspective for the prediction of hygrophilous plant dynamics and to facilitate the formulation of wetland management.

**Abstract:**

Flooding duration and sediment texture play vital roles in the growth and adaptation of wetland plants. However, there is a lack of research on the interactive effects of flooding duration and sediments on wetland plants. A two-factor experiment with flooding duration and sediment texture was designed in the study, involving three plant species commonly found in the Poyang Lake wetland (i.e., *Carex cinerascens*, *Phalaris arundinacea*, and *Polygonum criopolitanum*). Our findings were as follows: (i) Sediments play a crucial role in the growth and adaptation of hygrophilous plants, but they exhibited a weaker effect than flooding. (ii) Sediment texture mediates flooding to affect the stressing responses of wetland plant functional traits, including the leaf chlorophyll content, the plant height, and the number of leaves and ramets. (iii) Sediment texture forms interactive effects with flooding duration and directly influences hygrophilous plants. The results of this study help provide theoretical insights from a more scientific perspective for the prediction of hygrophilous plant dynamics and to facilitate the formulation of wetland management.

## 1. Introduction

Hygrophilous plants are crucial components of wetland ecosystems, which drive the energy flow and biogeochemical cycles of ecosystems [1,2]. Meanwhile, the survival and growth of wetland plants are also considerably influenced by the environmental conditions of wetland ecosystems, such as the hydrology and sediment dynamics [3,4,5]. Wetland plants have developed unique physiological and ecological adaptations to wetland environments [6,7]. However, in recent years, increased water resource utilization by humans, land reclamation, and climate change have profoundly changed the hydrology and sedimentation patterns of wetland ecosystems and accelerated their succession [8,9]. These environmental changes pose severe challenges to the growth and adaptation of hygrophilous plants.

Hydrologic stress on the growth of hygrophilous plants mainly occurs in the form of flooding [10]. Hygrophilous plants regulate various aspects of functional traits to acclimate to flooding stress [11]. Specifically, a plant would regulate chlorophyll content to respond to the limited light in a flooding environment and adjust stem height to escape the low oxygen level [12]. Plants can also adjust the above- and below-ground biomass allocation to adapt to submergence [13], and different hygrophilous plant species are deemed to have evolved differing tolerance thresholds to flooding [14]. In addition, the hydrological regime could shape the sediment conditions, such as water storage capacity and redox state [15,16,17], then indirectly affect the hygrophilous plants [18]. As for wetland sediments, they are key locations for material transformation by hygrophilous plants [19]. A sediment’s inherent properties, such as particle size and porosity from sediment texture, also exert a critical influence on the water and nutrient absorbability of plants and plant growth [19,20]. And the plant functional traits of seedlings to adapt to flooded soil conditions differed between sediment types, even among conspecific seedlings [21]. Therefore, flooding and sediment dynamics are two important research areas in hygrophilous plants. Several studies have examined the influence of hydrological factors on hygrophilous plants [22,23,24]; however, few studies have investigated how the texture of sedimentary soil affects hygrophilous plants [25]. Moreover, there is a lack of research on the interactive effects of hydrologic factors and sediments on hygrophilous plants, and the ecological mechanisms underlying these effects are unclear.

Poyang Lake wetland is an internationally important ecological functional area and is rich in biological resources [26]. The Poyang Lake wetland is a typical case in freshwater lake wetland research. As a Yangtze River-connected lake, Poyang Lake is characterized by seasonal water level changes. So, the plants on Poyang Lake’s beach always face seasonal flooding and are shaped to distribute in a zonal flooding pattern [27]. *Carex cinerascens* Kukenth., *Phalaris arundinacea* L., and *Polygonum criopolitanum* Hance are three plant species that have established large communities on the Poyang Lake floodplains [28]. But they tend to distribute on special sediments differing in texture under the same flooding conditions [26]. Sediment texture is considered to play a vital role in the growth and distribution of hygrophilous plants [28]. In recent years, the Poyang Lake wetland has become severely degraded, resulting in longer dry periods [29,30,31]. The fluctuating hydrology of Poyang Lake could mediate the transport and distribution of sediment to shape sediment types with different textures on the beach wetland [32]. The changing hydrological condition and sediment texture profoundly affect plant growth and change the structure of the wetland ecosystem [18,33]. Given this background, understanding how hygrophilous plants respond to these changes is important for properly managing the Poyang Lake wetland. Therefore, in the present study, we selected *C. cinerascens*, *P. arundinacea*, and *P. criopolitanum* as the study subjects. Then, five different flooding durations and five types of sediment texture were used in a two-factor experiment for the selected plant species to consider the following objective: evaluate the interactive effects of flooding and sediment texture on survival, biomass, and several functional plant traits. Our hypotheses were the following: (a) sediment textures involve the response of wetland plants to flooding stress; (b) plants show species-specific resistance to flooding stress in different sediment textural gradients, resulting in field plant distribution differences on sediment types. The results will enhance the prediction of the dynamics of hygrophilous plants and so facilitate the formulation of wetland management and restoration strategies.

## 2. Materials and Methods

### 2.1. Plant Materials

In late January 2021, we collected the broken branches of *P. criopolitanum* around Tang Yin Island (Duchang County, Jiujiang City, Jiangxi Province; 116°22′10″ E, 29°6′46″ N). *P. criopolitanum* is a prostrate growing annual herb that grows adventitious roots from branches to propagate ramets, which is a major way to produce plants [34]. So, we collected branches for *P. criopolitanum* to nurture homogeneous plants for experiment. The *C. cinerascens* and *P. arundinacea* plantlets were collected from the Nanji Mountain Wetland National Nature Reserve (116°15’29″ E, 28°53′35″ N) in early March 2021. The collected plants were also cultivated in a greenhouse (temperature 15–20 °C; watered daily) to obtain homogeneous plants. The cultivated plants of three plant species were selected in their intra-species similar sizes and conditions, respectively, and used as the experimental plant materials. 

### 2.2. Experimental Design

We mixed sand with the substrate soil in differing volumes to simulate the conditions of different sediment types distributed on the shoals and beaches in Poyang Lake region. The used sand was plain sand commonly used in construction with no nutrients, and we bought it from market. Before mixing, the sand was sieved to remove the gravel and stones in it and was homogenized (average particle size: 129.3 ± 0.49 μm). The screen diameter is 2 cm because matter larger than 2 cm is classified as either gravel or stone according to the American soil texture classification system [35]. The substrate soil was collected from Donghu (116°22′18″ E, 28°58′26″ N), a typical floodplain area of Poyang Lake. The substrate was characterized as a pretty fine texture (average particle size: 32.5 ± 0.61 μm) and corresponded to the sediments with no sand (Appendix A). The used substrate soil was also sieved to keep it homogenized before mixing. Five sediment types differing in texture were set in this study: 0% sand (per unit volume of sediments), 30% sand, 50% sand, 70% sand, and 100% sand. The sediment physicochemical properties change in terms of gradient with the set textures: the sediments with no sand possess best textures and highest nutrient-retention ability, while the sediments containing 100% sand feature roughest texture and lowest nutrient retention (Appendix A). The duration of flooding was also set to 5 levels—0, 15, 30, 60, and 90 days (d)—in a gradient that was within the flooding tolerance thresholds of the three plants. Therefore, plant mortality due to irreversible damage from flooding could be avoided during the experiment [36].

At the beginning of April 2021, the samples of the three plants were transplanted into uniform pots (diameter, 14.8 cm; height, 10 cm) for the experiment. The pots were, respectively, filled with the five types of sediments (sediment volume, 1 L). Then, the plants were cultivated in a moist environment for one week to adapt to the sediment conditions, following which the pots with plants were moved into a pool (water depth, 0.5 m) for flooding at the Hu Hsen-hsu Memorial Garden of Nanchang University (115°48′28″ E, 28°39′46″ N). Three replicates for 25 treatments, respectively, were set up for three plant species. In brief, a total of 225 pots were set in the experiment (5 flooding treatments × 5 sediment treatments × 3 plant species × 3 replicates) (Figure 1). The experiment ran from 28 April 2021 to 31 July 2021, and the plants outside the pool were watered regularly during this period.

### 2.3. Experimental Measurements

At the set nodes of flooding duration, as shown in Figure 1, we measured and recorded the following functional traits in the plants of each plant species: leaf chlorophyll content (LCC), plant height, and number of leaves and ramets. Then, we could obtain data for plant functional traits with the flooding duration series, respectively, on five sediments. After every measurement, the plants treated with the corresponding flooding duration were removed from the flooded pool and placed on the bank for recovery. At the same time, the other plants were moved into the pool to continue the flooding treatment. For example, after a group of plants had been subjected to flooding for 30 d, they were placed on the bank of the pool for recovery (along with the plants subjected to flooding treatments for 0 or 15 d). After 90 d, all plants in the different flooding treatment groups were subjected to the preset flooding duration. We recorded the survival of plants in each treatment group and harvested them on the seventh day after the end of the flooding period. The above-ground and below-ground parts of plants were rinsed with tap water, separated, dried at 60 °C for 48 h, and weighed. With this method, we obtained the above-and below-ground biomass and total biomass of the plants of the three species used in this study.

### 2.4. Data Analysis

All statistical analyses and data visualization were performed in R statistical software (version 4.0.4) [37]. To evaluate the survival of the three hygrophilous plant species under the interactive effects of flooding duration and sediment texture, the surviving plants in each experimental treatment were counted at the end of the experiment. To test the effects of sediment texture on plant adaptation to flooding stress, regression analysis was used to analyze the data for plant functional traits with flooding duration on five sediments. Regression models commonly used, including linear regression, logistic regression, and polynomial regression, were applied to fit the data [38]. We measured the goodness of fit by using R^2^adj and selected the optimal regression models. The effects of flooding duration and sediment texture on plant growth were evaluated with linear mixed effects models (LMMs) with flooding duration, sand content of the sediments, and their interaction as fixed factors and plant species as a random factor (y ~ flooding × sand content of the sediments + (1|species)) [39]. Plant biomass was used as a measure of comprehensive plant growth [40], and the total, above-ground, and below-ground biomass and ratio of above- and below-ground biomass (ABR) were used as response variables in the LMMs. We also tested an alternative model that did not consider the interaction between flooding duration and sediment texture (y ~ flooding + sand content of the sediments + (1|species)). However, this model had lower Akaike information criterion (AIC) and Bayesian information criterion (BIC) values and a higher log-likelihood (logLik) [41]. Thus, the model that included the interaction of the two experimental factors as a fixed effect provided a better fit to our data. We calculated the P-values from the LMMs using Type II Wald chi-square tests [42]. The effect sizes of the treatments and treatment interaction were represented by the standardized regression coefficients in the LMMs, which reflected the directions and magnitudes of the treatment effects [39].

We used the following R packages for the above analyses: HH [43], car [44], multcomp [45], lme4 [46], glmm.hp [47], and ggplot2 [48]. HH, car, multcomp, and lme4 were used for curve fitting and regression analysis; glmm.hp was used to calculate the effect sizes of fixed factors in the LMMs; and ggplot2 was used to visualize the results.

## 3. Results

### 3.1. Effects of Flooding Duration and Sediment Texture on the Survival of Hygrophilous Plants

The survival data of the three hygrophilous plant species in each treatment group were analyzed (Figure 2). The plants of *C. cinerascens* grew well in each treatment group. When the duration of experimental flooding was >30 d, there was a decline in the survival rates of *P. arundinacea* plants growing in sediments with mixed sand. After flooding for 90 d, the survival rates of *P. arundinacea* decreased with the increasing sand content of the sediments, and all individual plants growing on the sediments composed entirely of sand died. For flooding durations of up to 60 d, all *P. criopolitanum* plants survived on the five types of sediment. However, during flooding for 90 d, the plants growing on all five sediment types died to varying degrees. The survival rate of *P. criopolitanum* was lower on sediments with no sand than on sediments with mixed sand. However, all *P. criopolitanum* plants died when the sand content of the sediment was 100%. 

### 3.2. Effects of Flooding Duration and Sediment Texture on the Functional Traits of Hygrophilous Plants

#### 3.2.1. Leaf Chlorophyll Content

In this study, the LCC for all three plant species growing on all five sediment textures showed a significant linear decrease with an increase in flooding duration (Figure 3). Especially for the *P. criopolitanum,* the plant LCC on five sediments all decreased to close to 0 at 60 days of inundation. 

#### 3.2.2. Plant Height and Number of Leaves

In most experimental groups with different sediments, the plant height tended to increase and then decrease with an increase in flooding duration (Figure 4). The plant height of *C. cinerascens* declined consistently with flooding in the sediment with 100% sand (by volume, same as below). The plant height of *P. criopolitanum* was shorter in the sediment with 100% sand than in the other sediment treatments. The leaf numbers of *C. cinerascens* and *P. arundinacea* plants exhibited no significant patterns with changes in flooding duration or sediment texture (*p* > 0.05) (Figure 5). However, the leaf number of *P. criopolitanum* plants decreased linearly with flooding on all sediment types (Figure 5).

#### 3.2.3. Number of Ramets

The number of ramets in *C. cinerascens* plants increased linearly with flooding on all five sediment treatments, and the increase was greatest on sediments with no sand (Figure 6). The number of ramets was low in both *P. arundinacea* and *P. criopolitanum* plants and showed no consistent pattern in response to flooding duration on most of the experimental sediments (Figure 6). The number of ramets increased slightly with flooding in *P. arundinacea* and *P. criopolitanum* plants growing on sediments with no sand and mixed with 70% sand, respectively.

### 3.3. Effects of Flooding Duration and Sediment Texture on the Biomass of Three Hygrophilous Plants

To test the main and interaction effects of flooding duration and sediment texture on plant biomass, we performed LMMs and compared the effect sizes of fixed factors. Flooding duration and the sand content of the sediments had negative main effects on the total, above-ground, and below-ground biomass of the three species (*p* < 0.001), whereas their interaction had a positive effect on plant biomass (*p* < 0.001). Only flooding duration had a significant main effect on ABR (*p* < 0.001). With increasing flooding duration, all three species tended to allocate the biomass to above-ground parts (Table 1).

The two environmental factors had considerable interaction effects (flooding duration × sediments) on plant biomass in all three species (Figure 7). The total biomass of *C. cinerascens* plants decreased significantly with an increasing sand content of the sediments when the plants experienced flooding for <30 d (*p* < 0.05). For a flooding duration of ≥60 d, the total biomass of plants was higher on sediments mixed with a >50% volume of sand. The total biomass of the other hygrophilous plant species also exhibited similar patterns. Under non-flooding conditions, the total biomass of *P. arundinacea* and *P. criopolitanum* plants decreased significantly with the increasing sand content of the sediment. The biomass of these plants was significantly higher on sediments with no sand than on the other sediments (*p* < 0.05). However, when the plants experienced flooding for >30 d, the total biomass of *P. arundinacea* plants was highest in the sediment mixed with 50% sand, and that of *P. criopolitanum* plants was highest in the sediments mixed with 70% sand.

## 4. Discussion

### 4.1. The Texture of Sediments Plays a Crucial Role in Shaping Functional Traits of Hygrophilous Plants Respond to Flooding

Hygrophilous plants are known to adapt to environmental stresses by altering their morphological and physiological states [49]. In our experiment, we found that the responsiveness of the four functional traits to flooding stress varied with sediment texture. For the LCC, the harm from flooding was obvious; the LCC of three species of plants all showed a rapidly straight downward trend, and the decreasing rate differed in sediment treatments. The plant heights of three species first increased and then decreased in the flooding process, and height responsiveness also changed with the sand content of sediments. The response patterns in the flooding process of the three species of plant leaf number and ramet number seemed not as obvious as the above two traits, but some differences were exhibited in the responsiveness for five sediment textures. Accordingly, sediment texture produces vital impacts on a plant’s morphological and physiological responses in the flooding process. 

Chlorophyll, an essential pigment that facilitates photosynthesis, is particularly important for the physiological adaptation of plants to the environment [50]. In the flooding process, low levels of dissolved oxygen in water bodies result in higher levels of accumulation of toxic gases (such as CH_4_, H_2_S, and NH_3_) around the plant roots, which can injure the plant’s physiological activities and cause rapid decomposition of the leaf chlorophyll content (LCC) [51,52]. And the experimental sediments with distinctive soil textures differed in their gaseous cycles [53], which typically involve the interchange of gases in sediments. Sediments mixed with adequate amounts of sand showed improved soil aeration than sediments with no sand [53], which mitigated the accumulation of toxic gases in the sediments and relieved the harm to plants. In this study, there was a relatively mild decline in LCC in *C. cinerascens* and *P. arundinacea* plants growing on sediments mixed with 50% and 30% sand (by volume), respectively (Figure 3). Moreover, the sediment nutrient-retaining properties are closely related to texture [54]. Sediments with clay hold nutrients more effectively than sand due to their smaller particle size [55]. Thus, in our study, the nutrient-retaining ability decreased with the increasing sand content of sediments. Sediment nutrients are considered a critical element affecting the acclimation of hygrophilous plants to environmental stresses [56]. Low nutrient retention would restrict the resistance of plants to flooding [57]. This may explain the fastest declining rate of plant LCC for *C. cinerascens* and *P. arundinacea* on sediments with a 100% volume of sand.

LCC is linked closely to plant photosynthesis, which is considered the primary energy source for growth [50]. The decline of plant LCC in the flooding process could limit the energy supply and inhibit plant growth. In the study, that energy limitation was reflected in the decrease of plant height in three hygrophilous plant species after periods of flooding. The plant heights of *C. cinerascens* and *P. arundinacea* peaked after approximately 30 d of inundation, while that of *P. criopolitanum* peaked after approximately 45 d of inundation (Figure 4). The differing plant height changing rates in five sediment textures may be associated with the differing LCC decline rates in sediment treatments. Leaf number is another indicator of above-ground growth in plants [12]. When facing energy and resource limitations in flooding, some growth-related trade-offs existed between the plant height and leaf number. Hygrophilous plants are accustomed to escaping flooding stress through stem elongation [14], which requires increased cell division and is a very energy- and resource-intensive process [10]. This means that little energy and resource would be used for developing new leaves. Thus, in the experiment, the leaf number of plants growing on different sediments changed very little (*C. cinerascens* and *P. arundinacea*) or decreased with flooding stress (*P. criopolitanum*). 

Most hygrophilous plant species feature clonal propagation and response to environmental stress by the birth of ramets. The number of ramets for hygrophilous plants could reflect the environmental effects [58]. The differing responsiveness of ramet number in the study could be explained by a combination of direct impacts from sediment treatments and indirect impacts from differing physiological characteristic responses (LCC) in differing sediment textures. Additionally, ramets develop from adventitious roots on the stem [58], and the cells in them are loosely arranged [59]. The structure is beneficial for plant survival in the anoxic environment caused by flooding. In the study, the ramet number of *C. cinerascens* plants exhibited a significant linear increase with flooding on all five sediments, and the plant survival rate for *C. cinerascens* was higher than the other two species under experimental conditions. From this perspective, *C. cinerascens* possesses better flooding resistance than *P. arundinacea* and *P. criopolitanum*. 

### 4.2. Flooding Duration and Sediment Texture Interact to Affect Wetland Plant Biomass Accumulation

Plant biomass is a comprehensive indicator of plant responses that are easily and inexpensively measured and are therefore used as important plant traits as reported elsewhere [40,60]. In this study, the biomass of three plant species was significantly influenced by flooding duration and the texture of the sediments, indicating that these factors play key roles in regulating the growth of hygrophilous plants. Flooding exerted a significant negative effect on plant biomass. Under fully flooded conditions, the growth of non-aquatic plants is inhibited, and the accumulation of plant biomass remains at a standstill [61]. Until plants survive flooding, biomass accumulation occurs with the recovery of plant growth [62]. The longer the duration of flooding, the shorter the time period available for plants to regrow and recover in our experiment. Consequently, the biomass of plants subjected to ≤30 d of flooding was significantly higher than that of plants subjected to longer inundation of flooding treatments.

Based on our results, the plant biomass almost decreased with an increasing sand content of the set sediments following a ≤15 d of experimental flooding duration (*p* < 0.05), whereas plant biomass for three species all differed insignificantly among sediments following a ≥30d of experimental flooding duration (Figure 7). This could be attributed to the differing adaptation characteristics of plant functional traits on five sediment textures [63,64]. For three selected plant species, the negative responses of functional traits all seemed weaker on the sediments mixed with a medium volume of sand than other sediments, leading to lower levels of plant growth inhibition under flooding. This compensated for the accumulation of plant biomass [65] and compressed differences in plant biomass with low sand-content sediments. But when the plants recover from the flooded environment, the inhibition weak with the increasing period for plant recovery growth [66], and the final plant biomass would be predictably higher on the sediments with the excellent nutrient-retaining ability (mixed ≤30% sand in our experiment). Therefore, flooding duration and the texture of sediments interacted to exert a slightly positive effect on plant biomass overall, even though the two fixed factors had negative main effects on biomass.

In the study, three wetland plants showed similar response patterns to experimental flooding and sediment factors. The plants’ similar response patterns disprove our hypothesis b. But what causes the three selected plants to feature different distributions for sediment types in the Poyang Lake wetland? Indeed, some studies show that environmental factors control plant interspecific interactions and shape plant distribution. This suggests that sediment texture might influence hygrophilous plant distribution under flooding stress by mediating plant interspecific activity rather than growth adaptation. The underlying mechanism requires further experimental studies.

## 5. Conclusions

In conclusion, the results highlight that sediment texture plays a crucial role in the growth and adaptation of hygrophilous plants, although it exhibited a weaker effect than flooding duration. Additionally, sediment texture mediates flooding to affect plant stressing response and survival. Moreover, sediment texture forms interactive effects with flooding duration and has direct influences on hygrophilous plants. Thus, at the present time, environmental changes in lake wetlands are taking place, and the roles played by sediment texture, a neglected environmental element, require attention.

## Figures and Tables

**Figure 1 biology-12-00944-f001:**
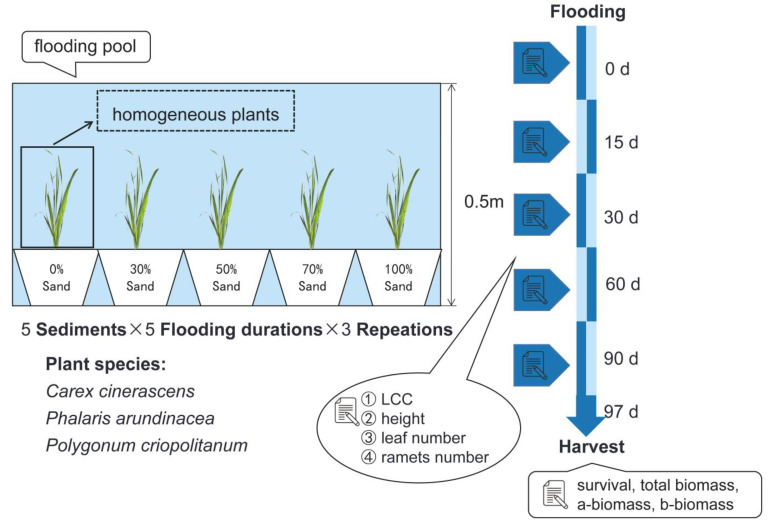
A schematic map of the two-variables experiment.

**Figure 2 biology-12-00944-f002:**
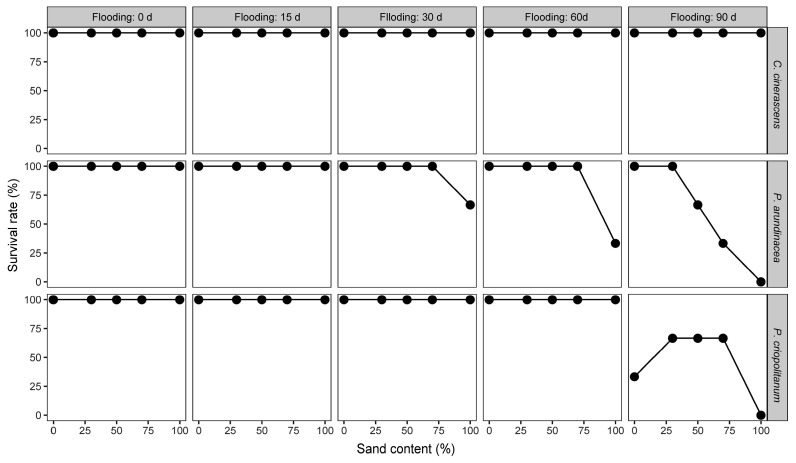
Effects of flooding duration and the sand content of sediments on the survival of three hygrophilous plant species. Sand content (%) indicates the proportion of sand mixed in the experimental sediments.

**Figure 3 biology-12-00944-f003:**
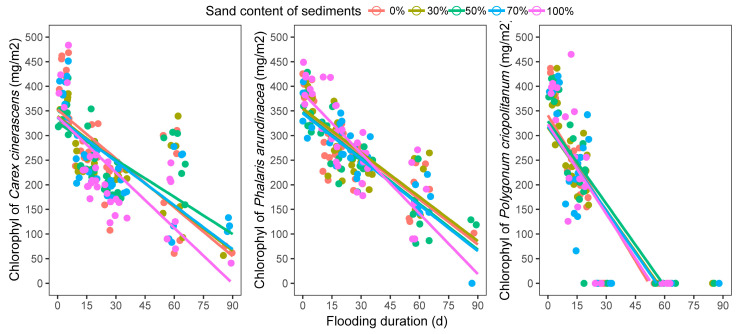
The response of plants’ leaf chlorophyllton content five sediments to the flooding.

**Figure 4 biology-12-00944-f004:**
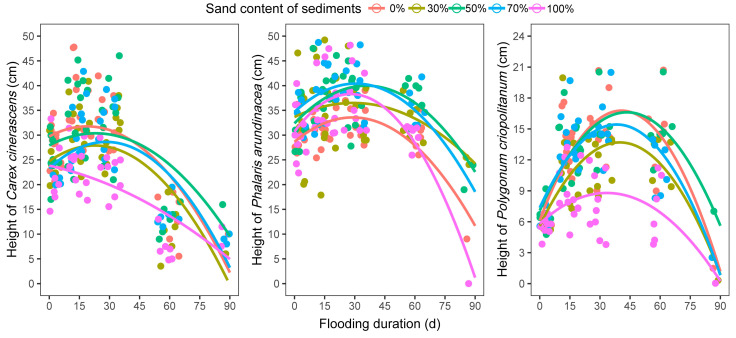
The response of plants’ height on five sediments to the flooding.

**Figure 5 biology-12-00944-f005:**
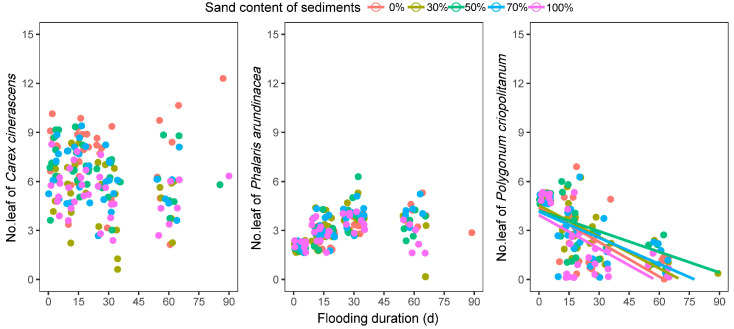
The response of seedlings’ leaf number on five sediments to the flooding. Significant (*p* < 0.05) fitted regressions were indicated by lines.

**Figure 6 biology-12-00944-f006:**
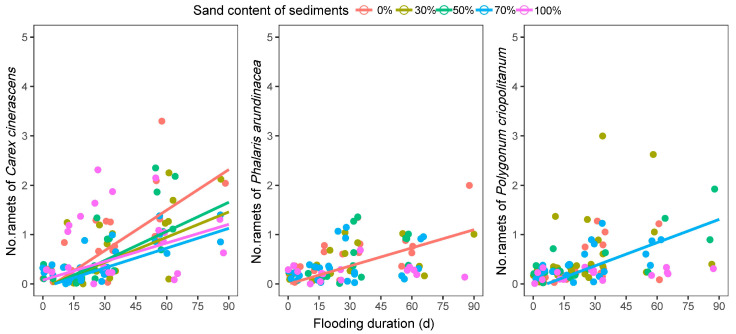
The response of seedlings’ ramets number on five sediments to the flooding. Significant (*p* < 0.05) fitted regressions were indicated by lines.

**Figure 7 biology-12-00944-f007:**
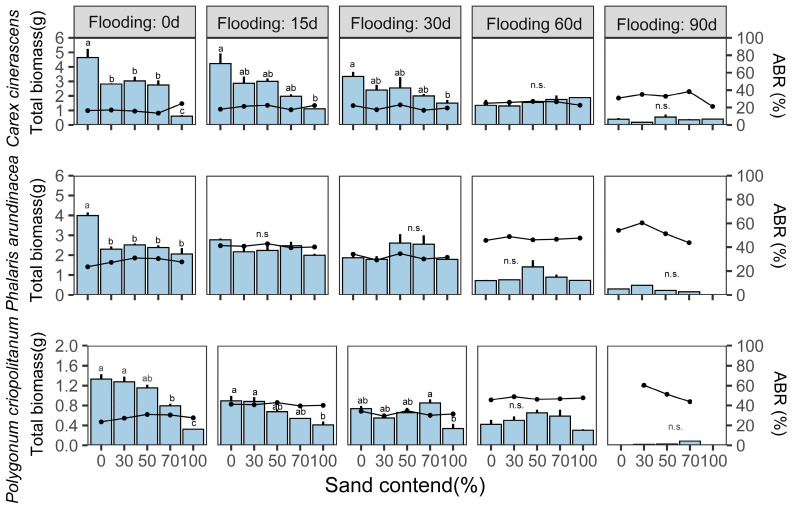
Changes in the total biomass and above-ground/below-ground biomass ratio (ABR) of three hygrophilous plant species in response to the interactive effects of flooding duration × sand content of the sediments. The bar charts show the total biomass, and the line charts show the ABR. Bars with differing lower-case letters indicate significantly different based on Tukey’s tests at the 5% level; n.s., *p* > 0.05.

**Table 1 biology-12-00944-t001:** Effect sizes of the three fixed factors in the linear mixed effects models. Sediments indicate the sediment texture and are reflected by sand content in the experimental sediments in the study; flooding indicates the duration of flooding; *** *p* < 0.001.

y	Fixed Factor		Random Factor
	Sediments	Flooding	Species
Total biomass	−0.50 ***	−0.93 ***	0.47 ***
Above-ground biomass	−0.48 ***	−0.89 ***	0.38 ***
Below-ground biomass	−0.46 ***	−0.86 ***	0.45 ***
ABR	−0.03	0.40 ***	−0.16

## Data Availability

All data are available in this article.

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
