# Peer review of "Interactive Effects of Flooding Duration and Sediment Texture on the Growth and Adaptation of Three Plant Species in the Poyang Lake Wetland"

_biology, 2023, doi:10.3390/biology12070944_

Round 1

Reviewer 1 Report

This is a very interesting paper, with clear results. I like the experimental set up. Introduction is informative and methods and results are well presented and sound. Some advises for improvement remain:

Objectives are not formulated well: Combine 1 and 3 by: Determine the interactive effects of flooding and sediment texture on survival, biomass and several functional plant traits of three wetland species. Objective 2 is formulated poorly and not understandable.

Align the set up of Fig 2 with Fig 2-6. So change the presentation of Fig 2 inot similar to the other figures.

Avoid over-interpretation. F.i. I do not see a difference between the chlorophyl trends in Fig 3. between the texture classes, whereas you spend quite some words on the slopes in line 196-202.    

I would like to see more emphasis on the differences between the 3 species. You have a two factor experiment but included three species. This is not very prominent in your manuscript. Stress this in the legend of Fig 1, first of all. And include in Abstract. Now no mention of interspecific differences, in Results you mention differences in response among species, but no implications are drawn in Discussion nor mentioned in the Conclusions.

In some instances it is hard to follow exactly what you mean. F.i. line 23, line 367 and 369 What do you mean with conducive ?, line 27/28 and line  371/372 What are the theoretical insights? Elaborate on this. Be explicit always, use clear and transparent language and avoid vague or general formulations.

Furthermore I miss the management implications and translation of results for application in Lake Poyang wetland. In the Introd. you stress the importance for the wetland, but nowhere you come back to the importance for the wetland. What could the managers learn?

References are too much biased towards Chinese references. Refer more frequently to relevant scientific papers form other continents . And replace part of the Chinese papers you refer to with others. This would make your manuscript much more interesting and potentially relevant for an international audience.

Objectives are not formulated well: Combine 1 and 3 by: Determine the interactive effects of flooding and sediment texture on survival, biomass and several functional plant traits of three wetland species. Objective 2 is formulated poorly and not understandable.

In some other instances it is hard to follow exactly what you mean. F.i. line 23, line 367 and 369 What do you mean with conducive ?, line 27/28 and line  371/372 What are the theoretical insights? Elaborate on this. Be explicit always, use clear and transparent language and avoid vague or general formulations.

English is in quite many instances poorly formulated and the ms contains typos.   

Reviewer 2 Report

MS Title: Interactive effects of flooding duration and sediment texture on  the growth and adaptation of three plant species in the Poyang  Lake wetland

The manuscript presents intriguing findings regarding plant-soil feedback and merits publication in the Biology. However, there are suggestions to enhance the manuscript and stimulate further improvement.

Introductions

The overall introduction lacks strength and completeness. The study hypotheses have not been adequately introduced. I kindly request you to present your study hypotheses and provide a thorough background explanation for each hypothesis

At some point in the introductions please introduce your plant traits of interest, may be the following literatures can be helpful;

De Battisti, D., Fowler, M. S., Jenkins, S. R., Skov, M. W., Bouma, T. J., Neyland, P. J., & Griffin, J. N. (2020). Multiple trait dimensions mediate stress gradient effects on plant biomass allocation, with implications for coastal ecosystem services. Journal of Ecology108(4), 1227-1240.

Bornette, G., & Puijalon, S. (2011). Response of aquatic plants to abiotic factors: A reviewAquatic Sciences731– 14https://doi.org/10.1007/s00027-010-0162-7

Puchkoff, A. L., & Lawrence, B. A. (2022). Experimental sediment addition in salt-marsh management: Plant-soil carbon dynamics in southern New England. Ecological Engineering175, 106495.

L 56-57: Well, then provide here what these studies found regarding the sediment textures and plant traits ??

L 61-62,  it is an experimental investigation , then why Poyang lake emphasize ? it can be a part of the materials and methods

Materials and methods:

L 99-103: what is the definition of the sand in your experiments (e.g., particle size etc)? Please provide some details ?

L108: What is mean by pretty fine texture (provide details of particle size).

L351-353: Not clear provide detail mechanisms how flooding can affect the biomass productions

Results

Please provide Table for LMER models, current description is not easy to understand.

Discussion

Incorporate your hypothesis into the discussion by examining how your findings support or refute it, and elucidate the underlying mechanisms drawing upon existing literature.

L335:the statement can be improved and needed to be supported by the appropriate literature for example: 

“Plant biomass is a comprehensive indicator of plant responses to the ecological stimuli and responses of the biomass is easy and inexpensive to measure and are therefore used as an important plant traits as reported elsewhere (Heminway & Wilcox 2022; De Battisti et al. 2020; Billah et al. 2022)”.

Heminway, A. W., & Wilcox, D. A. (2022). Response of Typha to phosphorus, hydrology, and land use in Lake Ontario coastal wetlands and a companion greenhouse study. Wetlands Ecology and Management, 1-14.

Billah, M. M., Bhuiyan, M. K. A., Islam, M. A., Das, J., & Hoque, A. R. (2022). Salt marsh restoration: an overview of techniques and success indicators. Environmental Science and Pollution Research29(11), 15347-15363.

Reviewer 3 Report

Dear authors, the work is very interesting. I have included some notes in the text. Pay attention to the scientific nomenclature of the species.

The level of English seems good, clear, understandable and fluent.

Round 2

Reviewer 2 Report

Dear Editor,

Thank you for inviting me to review the manuscript. I appreciate the improvements that have been made.  I believe the manuscript is now suitable for publication. However, I would like to suggest a few changes that the authors should incorporate during the proofreading stage.

Firstly, in lines 80-85, the statement regarding hypothesis (b) is unclear. It appears that the authors intended to convey a particular idea, but it is not effectively communicated. I recommend revising this section to enhance its clarity.

is it the authors wanted to express” There would be species specific differences in plant traits due to inundation periods in different sediment textural gradients”?

Secondly, in lines 351-361, the content discussed falls within the realm of the discussion rather than the conclusions. I suggest relocating this portion accordingly to ensure the manuscript adheres to the appropriate structure.

Overall, I believe these adjustments will further improve the manuscript and contribute to its overall quality.

Sincerely,

Reviewer 2#

see the above comment 
